# Cognitive Effects of a Cognitive Stimulation Programme on Trained Domains in Older Adults with Subjective Memory Complaints: Randomised Controlled Trial

**DOI:** 10.3390/ijerph20043636

**Published:** 2023-02-18

**Authors:** Isabel Gómez-Soria, Juan Nicolás Cuenca-Zaldívar, Beatriz Rodriguez-Roca, Ana Belén Subirón-Valera, Carlos Salavera, Yolanda Marcén-Román, Elena Andrade-Gómez, Estela Calatayud

**Affiliations:** 1Faculty of Health Sciences, University of Zaragoza, 50009 Zaragoza, Spain; 2Health Research Institute of Aragon, 50009 Zaragoza, Spain; 3Grupo de Investigación en Fisioterapia y Dolor, Departamento de Enfermería y Fisioterapia, Facultad de Medicina y Ciencias de la Salud, Universidad de Alcalá, 28801 Alcalá de Henares, Spain; 4Research Group in Nursing and Health Care, Puerta de Hierro Health Research Institute-Segovia de Arana (IDIPHISA), 28222 Majadahonda, Spain; 5Physical Therapy Unit, Primary Health Care Center “El Abajón”, Las Rozas de Madrid, 28231 Madrid, Spain; 6Department of Psychology and Sociology, Education Faculty, University of Zaragoza, 50009 Zaragoza, Spain; 7Department of Nursing, University of La Rioja, La Rioja, 26004 Logroño, Spain

**Keywords:** subjective memory complaints, global cognition, orientation, short-term memory, mini-examen cognoscitive

## Abstract

Age-related subjective memory complaints (SMC) are a common concern among older adults. However, little is known about the effects of cognitive stimulation (CS) interventions on subjective memory complaints. The aim of this study was to analyse the effectiveness of a CS programme on global cognition and cognitive functions of older adults with SMC. A randomised clinical trial was conducted on older adults with SMC, including 308 participants ≥65 years of age assessed 6 and 12 months after the intervention. The assessment instrument was the Spanish version of the Mini-Mental State Examination (MEC-35), and all domains of the instrument were assessed. For statistical analysis, the data were analysed using robust ANOVA with means truncated at 20% utilising a two-way repeated measures model, with between (groups) and within (measurements) factors. In post hoc tests, a Wilcoxon signed-rank test of exact permutations between groups and Bonferroni correction were applied. In post hoc between-group tests, significant differences were found: (1) post-treatment in MEC-35, temporal orientation, short-term memory (STM), global language and praxis, and language and praxis (*p* ≤ 0.005); (2) at 6 months in MEC-35, global orientation, temporal orientation, and STM (*p* = 0.005); (3) at 12 months in MEC-35, global orientation, temporal orientation, STM, global language and praxis, and language (*p* = 0.005). This study shows benefits in global cognition and orientation, temporal orientation, STM, and language in older adults with SMC.

## 1. Introduction

Age-related subjective memory complaints (SMC) are a common concern among older adults [1,2]. In general, most studies agree that the prevalence of SMC in older adults is between 25% and 50% [3].

SMC are defined as types of complaints made by subjects regarding their cognition, but no clear impairment is found by objective psychometric tests [4]. Said form of complaints affect half of all people over 65 years of age [5].

SMC can usually be evidenced without the presence of objective cognitive impairment [2]. However, older adults with SMC have a higher prevalence that is defined by poorer overall cognitive function [2], poorer memory [6,7] (in particular verbal memory [8]), immediate memory [9,10,11], delayed recognition memory [9,10,12], short-term memory (STM) [11] (in particular a subsystem of STM, working memory [13]) spatial disorientation [14], poorer attention and processing speed [15,16], and poorer executive function [4,7,15].

SMC are not only related to ageing [3,17] but are also associated with depressive symptoms [2,7,14,18], anxiety, lower functional performance [7,19], and generally poorer perceived health [2].

Additionally, older age, female gender, low education level, and low social activity have been associated with SMC in older adults [15,20]. These complaints are also an important risk factor for future progression to mild cognitive impairment (MCI) or dementia [19,21,22,23].

There is, therefore, great interest in conducting studies based on non-pharmacological interventions in older adults with SMC in order to prevent MCI and dementia by reducing aggravation of losses in memory, executive function, and other cognitive functions, especially in women with low levels of education [7].

Cognitive stimulation (CS) is a non-pharmacological intervention defined as “participation in a series of group activities and discussions (usually in a group), aimed at the general improvement of cognitive and social functioning” [24]. In addition, they have the advantage of generating more interest and encouraging more active participation in older adults [25].

The literature suggests that the effects of CS are observed only in tasks and skills similar to those in which individuals have been trained. This implies that the training effect is usually specific to the stimulated cognitive domain [26,27] and that multi-domain training modules are more effective in generating cognitive and functional transfer in both the short- and long term compared to single-domain training modules [28].

It is clearly demonstrated that CS helps to counteract age-related cognitive decline [29], which could have implications for public health policies focused on promoting healthy cognitive ageing and design of prevention and intervention programmes for older adults with subjective memory complaints [30]. However, more studies need to be conducted on people with SMC to detail the effects of these interventions [29,31] in preventing future progression to dementia [32].

Our study hypothesises that older adults with SMC may improve cognitively at the global level and in some cognitive domains as a result of participating in the CS programme.

Therefore, the study aimed to analyse the effectiveness of a CS programme in older adults with SMC on global cognition and on specific trained domains in the short-, medium-, and long term.

## 2. Materials and Methods

### 2.1. Study Design

This research has been conducted through a randomised controlled trial (RCT) following Consort standards in 308 community-dwelling older adults with SMC.

Inclusion in the study was based on the following criteria: (1) aged ≥65 and (2) ≥24 score on the MEC-35 (validated Spanish version of the Mini-Mental State Examination (MMSE)), i.e., no cognitive impairment [33], and presentation of SMC. Individuals were excluded if they (1) had received CS in the last year; (2) were institutionalised; (3) obtained a Lawton–Brody index (L–B) ≥ 3; (4) reported more than 6 points on the abbreviated Goldberg anxiety scale; (5) reported ≥12 points on the abbreviated Yesavage depression questionnaire (GDS-15); (6) scored <60 points on the Barthel index (BI); (7) presented deafness; (8) presented blindness; (9) presented neuropsychiatric disorders; or (10) presented motor disturbances.

### 2.2. Selection of Participants

Participants were recruited at the San José Norte-Centro Health Centre in Zaragoza (Spain). Randomisation was performed using an opaque box in which participants’ file numbers were placed and an anonymous person extracted the selected numbers. The lead author verified the inclusion criteria of the participants. A total of 367 candidates were evaluated. The inclusion criteria were met by 308 participants, who were distributed into two groups: 131 in the intervention group (IG) and 177 in the control group (CG). The assessors and occupational therapists delivering the intervention were different. In addition, the assessors were blinded and were different in each of the assessments conducted. The protocol for the different phases of the study can be viewed in Figure 1. The flow diagram of the participants can be viewed in Figure 2.

### 2.3. Intervention

Ten group sessions of 45 min, one day a week, were conducted in 5 subgroups of 26–27 persons. In total, the intervention lasted two and a half months. Four trained occupational therapists carried out the intervention in four subgroups of 25–26 participants each using coloured notebooks of mental activation [34]. Each notebook allows training of the cognitive domains previously assessed according to MEC-35: temporal orientation, spatial orientation, immediate memory, attention, calculation, STM, language, and praxis. They also include occupational elements of the model of human occupation [35]: professions, interests, and roles that allow different levels of complexity to be expressed and increase personal satisfaction.

The difficulty of the exercises was adapted into three groups in consideration of the cognitive level measured with the MEC-35: yellow (MEC-35: 32–35 points), orange (MEC-35: 28–31 points), and red (MEC-35: 24–27 points). Each session included four parts: (a) orientation to reality: questions about date, time, and place; (b) explanation of the cognitive aspect on which each session was to focus; (c) individual work by each participant, in which 4 exercises for each cognitive aspect were performed; (d) group correction of the individual exercises.

### 2.4. Assessment Instruments

Different sociodemographic and clinical variables were examined.

The socio-demographic variables studied were age, gender, marital status, education level, physical occupational status, mental occupational status, and family living arrangements. In addition, education level was divided into two subgroups (primary/higher). An attempt was made to establish the most basic classification possible as this variable was not initially considered for the inference analysis of the results. Subdivision of physical occupational status and mental occupational status was made according to three levels: low, medium, and high for each based on the classification of Grotz [36]. The clinical variables assessed were hypertension, diabetes, hypercholesterolemia, obesity, and cerebrovascular accident.

Existence of subjective memory complaints was assessed with the question “Do you have complaints about your memory?” (dichotomous answer (yes/no)) [37,38].

The primary variable was Mini-Cognitive Examination (MEC-35) [39], which is considered one of the most widely used short cognitive tests to study cognitive capacities in primary care. It evaluates eight components: temporo-spatial orientation (10 points), fixation memory (3 points), attention (3 points), and calculation (5 points), short-term memory (3 points), language and praxis (11 points). Its sensitivity is 85–90% and specificity is 69%. With this questionnaire, global cognition and cognitive functions were evaluated. Scores below 24 points could indicate dementia [33]. Unlike MMSE, MEC-35 includes a three-digit series to repeat two similarity items in reverse order, and subtraction is performed three by three from 30, instead of 7 by 7 from 100, as in the version of Folstein et al. As the number of items increases, the maximum score in this version reaches 35 points compared to 30 in the original one [39].

We considered using the Spanish version of MMSE (MEC-35) to assess global cognition and to observe if there was any change in cognitive functions. Other authors suggest further investigation as to whether the overall MMSE assessment reveals areas of concern [40]. Gómez Gallego et al. mention that the MMSE enables rapid assessment of cognitive functions and evaluates the functions of different domains [41]. The data validity of the individual MEC items are also satisfactory (particularly with temporal orientation) [33]. In Spain, the adaptation of the MMSE carried out by Lobo et al. in 1979 [42], titled MEC, is commonly employed because some items of the original version by Folstein are difficult for patients of a low cultural level, which affects the scale’s discriminative capacity [39].

The tools used in the inclusion criteria are as follows:

#### Goldberg Anxiety Subscale

Anxiety was measured by the Goldberg anxiety subscale, which is a subscale of the Goldberg questionnaire, with nine dichotomous response items (yes/no). An independent score is awarded for each scale, with one point for an affirmative answer. The cut-off value is ≥4 for the anxiety subscale, which indicates “probable anxiety”. This scale has a specificity of 91% and a sensitivity of 86% [43].

#### Yesavage Geriatric Depression Scale 15-point Version

Level of depression was evaluated with the GDS-15, which is considered suitable for seniors in the community. Scores range from 0–15, with a total score > 5 interpreted as “probable depression”. Scoring higher than 12 would be indicative of severe depression. In older people, sensitivity is 71.8% and specificity is 78.2%, for a cut-off of 5 points [44].

#### The Barthel Index

The BI assesses the level of independence of 10 basic ADL (BADL) [45]. The maximum score for the BI is 100, where scores higher than 60 denote low dependence with ADL and scores below 20 demonstrate high dependence with ADL. Internal consistency was 0.90, with an inter-observer reliability Kappa index of between 0.47 and 1.00 and inter-observer reliability Kappa index between 0.84 and 0.97. Cronbach’s alpha was 0.90–0.9228 for the internal consistency evaluation [46].

#### The Lawton–Brody Scale

The L–B scale assesses degree of autonomy in eight IADL necessary for living independently in the community [45]. Scores range from 0–8 points. A score of 3 or less would be considered indicative of moderate dependence. Its sensitivity is 0.57 and its specificity is 0.92 [47]. The minimal important change of the Lawton IADL scale is around half a point. The certainty of this conclusion is reduced by variation across calculation methods [48].

The assessment process was carried out by eight occupational therapists who were blinded after receiving the corresponding training to ensure uniform application of the assessment instruments.

### 2.5. Sample Size

This study is a secondary analysis of aggregated data from two studies in which the sample size was already calculated and in which randomisation was conducted. The data of the present study are the result of merging these two previous studies, in which only participants with SMC have been selected. Both studies were approved by the Clinical Trials Ethics Committee of Aragon (CEICA) and registered in the clinical trials (see Ethical Considerations section). All patients in both studies signed an informed consent form and were given an information sheet.

The sample size for the study by Calatayud et al. (2020) [49] was calculated to be at least 97 persons per group. This size enables detection of difference of 1.5 points on the main variable with a power of 80% and a significance level of 5%, with an expected drop-out rate of up to 35%. The sample size of the study by Gomez-Soria et al. (2020) [50] was calculated in such a way that a 1.5-point increase in the MEC-35 could be detected with a significance level of 5% and a statistical power of 80% and assuming standard deviation of ≤2.5 points and a drop-out rate of 35%.

### 2.6. Statistical Analysis

Statistical analysis was performed using R Ver. 3.5.1. software (R Foundation for Statistical Computing, Institute for Statistics and Mathematics, Welthandelsplatz 1, 1020 Vienna, Austria). The significance level was set at *p* < 0.05.

Qualitative variables were described in absolute values and frequencies and quantitative variables were described with mean and standard deviation.

In each group, a Lilliefors-corrected Kolmogorov–Smirnov test was applied to assess the distribution of variables.

The presence of significant differences between the two groups in both age and gender was explored using a repeated measures linear mixed model and restricted maximum likelihood (REML) approach with each outcome variable. Subjects were modelled as a random effect and the group:time interaction as a fixed effect, including age and gender as covariates.

Due to the non-normal distribution of the outcome variables, they were analysed by robust ANOVA with means truncated at 20% using a two-way repeated measures model with between (groups) and within (time measurements). At the same time, for post hoc tests, we applied an exact permutations test between groups and the Wilcoxon signed-rank test within groups. In both cases, Bonferroni correction was applied.

The effect size was calculated with the η^2^_p_ (partial eta squared) statistic obtained by bootstrapping due to the non-normal distribution of the variables, which were defined as small (0.01–0.06), moderate (0.06–0.14), or large (>0.14). In the post hoc tests, the non-parametric r statistic was used as effect size, defined as <0.4 (small), 0.4–0.6 (moderate), and >0.6 (large).

## 3. Results

This study included 308 older adults with MEC-35 scores between 26 and 35 points; 66.05% (200) were women and 33.95% (108) were men. The mean age was 73.56 years. Table 1 shows the demographic and clinical characteristics of the participants. There were baseline differences between the two groups according to age and gender. Of the total number of participants, 131 belonged to the IG and 177 to the CG.

The ANOVA test of the linear mixed model shows that group–time interaction, adjusted for age and gender, still maintains significant differences (*p* < 0.05) in the global orientation, temporal orientation, and STM variables.

Significant differences (*p* < 0.05) in group–time interaction are tested in the MEC-35 regarding global orientation, temporal orientation, global language and praxis, language, and praxis variables with small and significant effect sizes. There are also significant differences (*p* < 0.05) in time main effect in MEC-35 variables attention and global calculation, attention, calculation, STM, language and global praxis, and language and in group main effect in MEC-35 variables global orientation, temporal orientation, STM, language and global praxis, language, and praxis (Table 2).

In the post hoc between-groups tests, differences are evident in MEC-35 variables (*p* ≤0.001), temporal orientation (*p* = 0.02), STM (*p* = 0.012), global language and praxis (*p* ≤ 0.001), language (*p* ≤ 0.001), and praxis (*p* ≤ 0.001) post-treatment; at 6 months in global variables (*p* < 0.001), global orientation (*p* ≤ 0.001), temporal orientation (*p* ≤ 0.001), and STM (*p* = 0.008), and at 12 months in global variables (*p* ≤ 0.001), global orientation (*p* = 0.004), temporal orientation (*p* = 0.008), STM (*p* = 0.004), global language and praxis (*p* ≤ 0.001), and language (*p* = 0.012). Consistently higher values are found in the IG compared to the CG and small and significant effect sizes (Table 3).

In the intra-group post hoc tests, statistically significant improvements are evident only in the IG in the variables of MEC-35 and STM at all times of measurement compared to the pre-treatment values (*p* < 0.001), as well as in STM at 12 months compared to post-treatment (*p* = 0.038). In global orientation and temporal orientation, improvements in IG are only evident at 6 months vs. pre-treatment with significant differences (*p*= 0.009 and *p* = 0.019, respectively). In language and praxis and in language, there is also improvement in IG at post-treatment vs. pre-treatment and at 12 months vs. pre-treatment (*p* ≤ 0.001) and also in language at 6 months vs. pre-treatment (*p* = 0.015). In attention and calculation, IG improves at post-treatment and at 6 months compared to pre-treatment (*p* = 0.001 and *p* = 0.003, respectively). In attention, both groups improved, with significant differences at post-treatment, 6 months, and 12 months compared to pre-treatment. In calculation, both groups worsen at post-treatment versus pre-treatment and at 12 months versus pre-treatment, but the CG worsens more; the CG also worsens at 6 months versus pre-treatment, with statistically significant differences being observed in all of them (Table 4).

## 4. Discussion

This study demonstrated that administration of a multi-domain CS programme adapted to pre-existing cognitive level (measured with the MEC-35) in older people with SMC resulted in global and specific cognitive improvements in some of the trained domains of the MEC-35, which are highly relevant for preventing MCI. These effects in the IG compared to the CG not only occur after the intervention but are maintained over time, both at follow-up I (6 months) and in the longer term at follow-up II (one year after the intervention).

In relation to the global cognitive effects obtained through our programme, such effects have also been recorded in other programmes in older adults with SMC. In those cases, the effects were obtained through multimodal and unimodal interventions [51] and also through CS, where assessment of these benefits was conducted using the MMSE [52]. In addition, some authors advised older adults with SMQ to reduce risk of cognitive decline through physical activity, cognitive stimulation, and a healthy diet [53,54]. The scientific explanation is that such interventions produce an increase in structural grey matter volume in brain regions encompassing the episodic memory network, with an expansion of cortical volume to a degree comparable to that of healthy training participants [55].

Therefore, these types of interventions could decrease the negative longitudinal association between people with SMC and their global cognition scores at 6 years [56]. However, it is necessary to adapt the stimulation to the pre-existing cognitive level of the older adult, as indicated by Gheysen et al. (2018) [57]. A sufficient cognitive challenge seems more important in obtaining overall cognitive effects than increasing the number of intervention sessions.

Regarding specific domains and follow-up times in our study, post-treatment improvements were found in MEC-35 variables temporal orientation, STM, language, and praxis. Improvements were found at 6 months in global and temporal orientation and STM and at 12 months in global and temporal orientation, STM, language, and praxis. Higher values were always found in the IG compared to the CG, as were small and significant effect sizes.

First, with regard to STM, we have verified the hypothesis initially proposed in our study. Indeed, we observe in the literature that STM is the most widely studied domain, in any form of intervention, and that which is most related to SMC. In our investigation, we have identified other studies that also report improvements in the cognitive domain of memory. Some studies have achieved these results after single-domain training that only includes training of memory strategies and meta-memory in their intervention [25,58,59,60], whilst other authors who administered CS for treatment of SMC also found improvements in memory [52]. These strategies, which have also been used in our intervention methodology (but in our case reinforced in each cognitive domain), emphasise learning and control over the cognitive processes of memory, helping older adults to understand aspects and processes of their memory [61,62]. This would produce an increase in cortical thickness of the prefrontal regions, which are related to metacognition [63], even with a significantly greater decrease in their SMC for everyday memory [64].

The effects of our programme on memory persisted following intervention at 6 and 12 months. Other RCT studies applying CS also achieve these effects post-treatment [58,65] at 6 months [58] and 9 months [32] and not only in patients with SMCs but also in those with a diagnosis of MCI. However, in this case, it was a prospective randomised study in Italy and a multimodal intervention that had the same number of sessions as in our study, with said sessions lasting longer than 90 min [66]. These medium- and long-term cognitive effects were achieved in different countries with different intervention modalities, but they were all conducted on older people with SMC and in medical centres. Thus, the RCT of Kang et al. (2021) [65] was a multidomain intervention using virtual reality in shorter sessions than ours and stimulation by levels. It employed the Korea-adapted version of MMSE, and the average age of its participants was very similar to ours. In contrast, the RCT of Frankenmolen et al. (2018) [58] used only memory strategies in a population younger than ours (8 years younger on average) in the Netherlands. Finally, the RCT of Kwok et al. (2013) [32] was conducted in Hong Kong, on a population of the same education level and a very similar mean age to ours, using a Loci multidomain methodology and the Chinese version of the MMSE.

Other modalities that do not refer to traditional interventions also obtain favourable improvements in memory. Thus, computerised cognitive training programmes [67,68,69,70], virtual reality [65,71], and multimodal programmes that combine CS with aerobic exercise not only obtain improvements in memory but these improvements are transferred to the cognitive domain of attention and logical reasoning [72] and executive functions [73,74]. In particular, virtual reality programmes had already shown benefits in older people with normal cognition who did not have SMC [75,76]. However, in our study, this transfer of attention did not occur, perhaps because it was a multi-domain programme but not a multimodal one. It should be noted that, in these cases, the average number of participants is lower and it is even a requirement that they have compulsory secondary education.

Moreover, cognitive stimulation relies on and improves cholinergic activity [77], which is known to be responsible for memory performance [78]. This aspect is fundamental in these patients, especially those evolving to MCI in which cholinergic dysfunction has been shown to already be impaired and potentially restored by drugs [79].

Regarding the cognitive language domain, we obtained improvement even though language skills decrease with age [80,81] and SMC patients have lower scores on fluency tests [10]. Other studies also find linguistic improvement but with multi-component programmes [52,66,82,83] compared to our multi-domain programme specifically focused on the language domain.

Temporal orientation is a less studied domain, perhaps because, in subjects without cognitive impairment, it is not affected, as opposed to groups with impairment, in which it is very evident [84]. However, supporting the intervention with calendars and writing the date on each of the activities performed has resulted in a positive benefit in this aspect, as in other studies [85]. Furthermore, it has been established that performance in cognitive tests, such as orientation, is related to frequency of SMC. Participants with no orientation lapses have an SMC frequency of 22.2% and subjects who fail all orientation items have a frequency of 93% [14].

We have also obtained positive results in praxis. It should be noted that other studies achieve these benefits with multimodal therapies combining physical and cognitive training [86], while, in our study, this was achieved with multi-domain training. It seemed important to us to highlight the importance of cognitive stimulation brought on by pencil and paper in our study, precisely to continue these practices that older adults used in their working lives and that may contribute significantly to cognitive/physical function after retirement [87]. However, a recent computer-based programme suggests that these programmes may be better suited to achieving the objective criteria of successful ageing than paper-and-pencil memory training programmes, but they note that this conclusion should be taken with caution as differences in age and education level may have influenced the results [88]. In this respect, a high percentage of people in our study had only primary school education.

In general, other studies show that the participants’ SMC predict a decline in language, whereas, if these complaints are noted by another informant, they would be more related to a decline in executive function and memory [56] and would predict cognitive and functional decline over 4 years [89].

Finally, considering that older people with SMC but who do not present objective complaints are twice as likely to develop dementia as individuals without SMC and that approximately 2.3% and 6.6% of older people with SMC will progress to dementia and MCI within one year [56,90,91,92], it is considered necessary to promote coping strategies in primary care. Thus, according to a recent systematic review, the non-pharmacological strategies most frequently advised by primary care physicians are increased physical activity, cognitive stimulation, diet, and social stimulation [31]. As a result, there is much interest in lifestyle approaches [93].

Single-question patient assessment of SMC, as assessed in our study, is considered an efficient objective tool to discriminate patients with dementia from healthy older adults in the community [94]. If informants are asked, executive function and temporal orientation, as well as memory, should be assessed, aspects that have been noted earlier in our study [95]. Further research is needed, however, on screening cognitive assessment in primary care to strengthen the current evidence, determine use of specialists [96], and maintain a gender perspective as men and women seem to show different meta-cognitive abilities in detecting and reporting changes in their memory [97].

Our study has some limitations to consider. First, we used the same tool for screening and selecting participants as we did for assessing the different subdomains. Second, we did not consider psychological aspects, such as anxiety and depression, that may have influenced the SMC. Finally, it must be noted that the effect of the intervention was not compared with a group of participants who did not present SMC.

It would be of great interest to conduct RCT on participants with SMC with strong samples that apply unimodal cognitive interventions, such as CS, or multimodal programmes, which may include CS and physical exercise, and evaluate their efficacy in this population.

## 5. Conclusions

The CS programme shows benefits in global cognition and in the specific domains of temporal orientation, STM, and language and praxis in older adults with SMC. These domains are highly relevant in preventing MCI in a population with a higher vulnerability to MCI due to the existence of such complaints.The multi-domain cognitive stimulation programme enables transfer of effective results in the trained areas except for immediate memory and attention, where other intervention modalities would need to be added.Paper and pencil tools are effective in improving the cognitive domains trained, especially in low educational environments.Primary care is the ideal place to detect these SMC and propose activities related to lifestyle or refer to specialists who can properly diagnose cognitive deficits and provide ongoing monitoring and evaluation.

## Figures and Tables

**Figure 1 ijerph-20-03636-f001:**
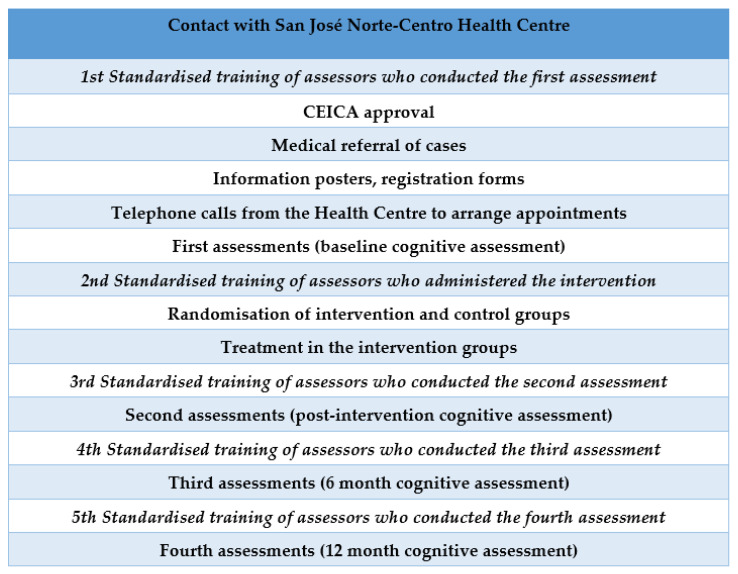
Protocol for the different phases of the study.

**Figure 2 ijerph-20-03636-f002:**
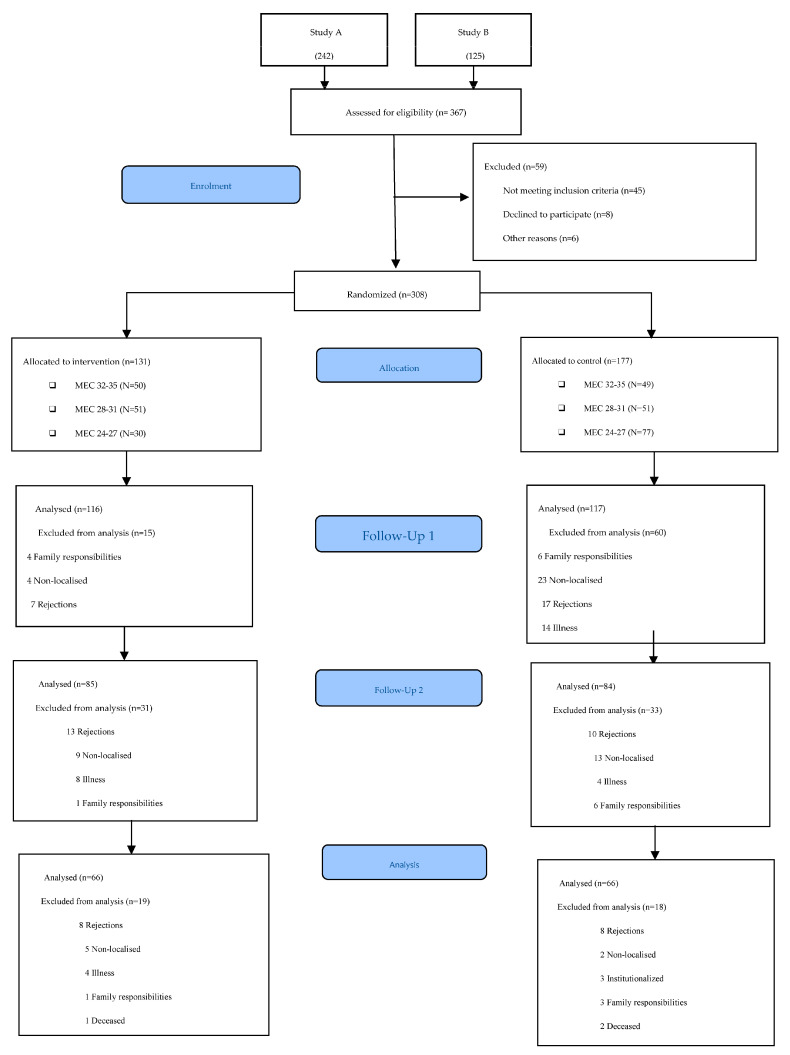
CONSORT 2010 flow diagram.

**Table 1 ijerph-20-03636-t001:** Demographic and clinical characteristics of participants.

		Total Sample	Intervention Group	Control Group	*p* Value ^1^
n		308	131	177	
Age *Mean (SD)*		74.56 ± 5.81	72.89 ± 5.54	74.23 ± 6.08	0.047
Gender, n (%)	Male	108 (36)	35 (26.7)	73 (41.2)	0.012
	Female	200 (64)	96 (73.3)	104 (58.8)	
Marital status, n (%)	Married	207 (67.2)	87 (66.4)	120 (67.8)	0.929
	Separated	10 (3.2)	4 (3.1)	6 (3.4)	
	Single	16 (5.2)	6 (4.6)	10 (5.6)	
	Widowed	75 (24.3)	34 (26.0)	41 (23.2)	
Level of education, n (%)	Primary education	235 (76.3)	104 (79.4)	131 (74.0)	0.336
	Secondary or higher education	73 (23.7)	27 (20.6)	46 (26.0)	
Physical occupation, n (%)	High	121 (50.4)	52 (39.7)	69 (39.0)	0.992
	Low	64 (20.8)	27 (20.6)	37 (20.9)	
	Medium	123 (39.8)	52 (39.7)	71 (40.1)	
Mental occupation, n (%)	High	17 (5.5)	5 (3.8)	12 (6.8)	0.248
	Low	178 (57.8)	72 (55.0)	106 (59.9)	
	Medium	113 (36.7)	54 (41.2)	59 (33.3)	
High blood pressure, n (%)	No	157 (51)	64 (48.9)	93 (52.5)	0.6
	Yes	151 (49)	67 (51.1)	84 (47.5)	
Diabetes, n (%)	No	266 (83.4)	116 (88.5)	150 (84.7)	0.427
	Yes	42 (13.6)	15 (11.5)	27 (15.3)	
Hypercholesterolemia, n (%)	No	187 (60.7)	84 (64.1)	103 (58.2)	0.35
	Yes	118 (38.3)	47 (35.9)	74 (41.8)	
Obesity, n (%)	No	267 (86.7)	113 (86.3)	154 (87.0)	0.983
	Yes	41 (13.3)	18 (13.7)	23 (13.0)	
Stroke, n (%)	No	288 (93.5)	122 (93.1)	166 (93.8)	>0.999
	Yes	20 (6.5)	9 (6.9)	11 (6.2)	

Data expressed as mean ± standard deviation or as absolute and relative values (%).^1^ Significant if *p* < 0.05.

**Table 2 ijerph-20-03636-t002:** Outcome variables.

	Intervention Group				Control Group							
	Pre-Treatment	Post-Treatment	2 Months	3 Months	Pre-Treatment	1 Week	2 Months	3 Months	*p* Value Time ^1^	*p* Value Group ^1^	*p* Value Group:Time ^1^	(95% CI)
MEC-35	29.824 ± 2.624	31.634 ± 2.395	32.071 ± 2.511	32.273 ± 2.402	29.915 ± 2.639	30.085 ± 3.167	30.357 ± 3.281	30.485 ± 2.78	<0.001	<0.001	0.002	0.008 (0.007, 0.044)
*Global orientation*	9.198 ± 1.033	9.431 ± 0.805	9.671 ± 0.714	9.545 ± 0.915	9.22 ± 0.949	9.085 ± 1.323	9.024 ± 1.242	8.939 ± 1.036	0.413	<0.001	0.001	0.004 (0.005, 0.037)
*Time orientation*	4.542 ± 0.715	4.681 ± 0.667	4.776 ± 0.624	4.667 ± 0.771	4.497 ± 0.762	4.376 ± 0.962	4.274 ± 1.034	4.212 ± 0.851	0.521	<0.001	0.022	0 (0.003, 0.033)
*Spatial orientation*	4.656 ± 0.617	4.733 ± 0.565	4.894 ± 0.31	4.879 ± 0.412	4.723 ± 0.581	4.709 ± 0.708	4.75 ± 0.557	4.727 ± 0.57	0.061	0.81	0.611	0.006 (0.001, 0.02)
*Global attention and calculation*	6.099 ± 1.626	6.578 ± 1.544	6.753 ± 1.55	6.606 ± 1.456	6.294 ± 1.557	6.453 ± 1.694	6.357 ± 1.655	6.5 ± 1.657	0.018	0.462	0.263	0.008 (0, 0.018)
*Attention*	1.672 ± 1.205	2.69 ± 1.398	2.765 ± 1.315	2.667 ± 1.293	1.853 ± 1.394	2.65 ± 1.522	2.643 ± 1.323	2.712 ± 1.465	<0.001	0.805	0.76	0.008 (0.001, 0.018)
*Calculation*	4.427 ± 0.985	3.888 ± 1.625	3.988 ± 1.651	3.924 ± 1.582	4.475 ± 0.905	3.803 ± 1.647	3.726 ± 1.689	3.788 ± 1.714	0.009	0.115	0.405	0.011 (0, 0.013)
*Short-term memory*	1.718 ± 0.987	2.216 ± 0.853	2.482 ± 0.796	2.606 ± 0.653	1.593 ± 1.078	1.838 ± 1.066	2.036 ± 1.035	2.076 ± 1.057	<0.001	<0.001	0.423	0.006 (0.001, 0.022)
*Language and global praxis*	9.809 ± 0.921	10.405 ± 0.78	10.165 ± 1.01	10.515 ± 0.707	9.808 ± 1.101	9.718 ± 1.105	9.94 ± 1.101	9.97 ± 0.96	<0.001	<0.001	<0.001	0.008 (0.007, 0.045)
*Language*	5.344 ± 0.721	5.75 ± 0.558	5.647 ± 0.685	5.833 ± 0.514	5.362 ± 0.801	5.427 ± 0.78	5.576 ± 0.746	5.5 ± 0.707	<0.001	<0.001	0.002	0 (0.003, 0.03)
*Praxis*	4.466 ± 0.683	4.664 ± 0.51	4.518 ± 0.648	4.682 ± 0.501	4.452 ± 0.648	4.299 ± 0.686	4.388 ± 0.638	4.47 ± 0.561	0.638	<0.001	0.045	0.001 (0.003, 0.034)

MEC-35: Lobo’s cognitive mini-examination. ^1^ ANOVA test *p* values for time (within measurement time in each group) and group (between both groups) main effects and group:time interaction; significant if *p* < 0.05.

**Table 3 ijerph-20-03636-t003:** Pairwise comparisons between groups.

	Pre-Treatment			Post-Treatment			6 Months			12 Months		
	Difference (95% CI)	*p* Value ^1^	r (95% CI)	Difference (95% CI)	*p* Value ^1^	r (95% CI)	Difference (95% CI)	*p* Value ^1^	r (95% CI)	Difference (95% CI)	*p* Value ^1^	r (95% CI)
MEC-35	0.186 (−0.387, 0.76)	>0.999	0.016 (0.001, 0.159)	1.571 (0.762, 2.379)	<0.001	0.218 (0.053, 0.319)	1.967 (0.767, 3.166)	<0.001	0.206 (0.098, 0.297)	1.611 (0.262, 2.96)	<0.001	0.213 (0.096, 0.328)
*Global orientation*	0.062 (−0.152, 0.277)	>0.999	0.008 (0.001, 0.121)	0.358 (0.065, 0.652)	0.068	0.104 (0.007, 0.224)	0.767 (0.439, 1.094)	<0.001	0.247 (0.144, 0.334)	0.694 (0.217, 1.172)	0.004	0.228 (0.129, 0.332)
*Temporal orientation*	0.079 (−0.086, 0.245)	>0.999	0.022 (0.002, 0.149)	0.34 (0.114, 0.565)	0.02	0.168 (0.065, 0.294)	0.617 (0.351, 0.882)	<0.001	0.237 (0.129, 0.337)	0.472 (0.089, 0.856)	0.008	0.208 (0.1, 0.314)
*Spatial orientation*	−0.017 (−0.147, 0.113)	>0.999	0.059 (0.001, 0.158)	0 (−0.178, 0.178)	>0.999	0.003 (0.004, 0.138)	0.15 (−0.014, 0.314)	0.168	0.094 (0.011, 0.212)	0.222 (0.039, 0.405)	0.468	0.118 (0.019, 0.233)
*Global attention and calculation*	−0.113 (−0.454, 0.228)	>0.999	0.052 (0.005, 0.17)	0.113 (−0.347, 0.573)	>0.999	0.026 (0.003, 0.158)	0.733 (0.101, 1.366)	0.46	0.096 (0.006, 0.208)	−0.194 (−1.054, 0.665)	>0.999	0.014 (0.002, 0.155)
*Attention*	−0.141 (−0.428, 0.146)	0.96	0.056 (0.001, 0.193)	−0.17 (−0.612, 0.272)	>0.999	0.012 (0.003, 0.143)	0.017 (−0.486, 0.519)	>0.999	0.019 (0, 0.145)	−0.528 (−1.258, 0.202)	>0.999	0.015 (0.002, 0.142)
*Calculation*	−0.006 (−0.202, 0.19)	>0.999	0.011 (0.002, 0.15)	0.283 (−0.203, 0.769)	>0.999	0.022 (0.002, 0.112)	0.7 (0.004, 1.396)	>0.999	0.066 (0.002, 0.188)	0.306 (−0.58, 1.191)	>0.999	0.013 (0.005, 0.131)
*Short-term memory*	0.209 (−0.013, 0.431)	>0.999	0.053 (0.002, 0.148)	0.349 (0.085, 0.614)	0.012	0.15 (0.03, 0.264)	0.383 (0.04, 0.727)	0.008	0.169 (0.053, 0.267)	0.5 (0.116, 0.884)	0.004	0.171 (0.073, 0.302)
*Language and global praxis*	0.028 (−0.185, 0.242)	>0.999	0.034 (0.003, 0.156)	0.736 (0.47, 1.002)	<0.001	0.289 (0.19, 0.373)	0.083 (−0.381, 0.548)	0.768	0.089 (0.006, 0.226)	0.611 (0.295, 0.927)	<0.001	0.212 (0.099, 0.315)
*Language*	−0.006 (−0.164, 0.153)	>0.999	0.035 (0.001, 0.149)	0.33 (0.149, 0.512)	<0.001	0.211 (0.105, 0.311)	0 (−0.317, 0.317)	>0.999	0.038 (0.002, 0.16)	0.333 (0.091, 0.576)	0.012	0.2 (0.087, 0.299)
*Praxis*	0.028 (−0.109, 0.166)	>0.999	0.015 (0.002, 0.144)	0.406 (0.232, 0.58)	<0.001	0.243 (0.102, 0.355)	0.049 (−0.191, 0.29)	0.936	0.086 (0.006, 0.18)	0.278 (0.04, 0.515)	0.14	0.131 (0.021, 0.256)

95% CI: 95% confidence interval; MEC-35: Lobo’s cognitive mini-examination. ^1^ Significant if *p* < 0.05.

**Table 4 ijerph-20-03636-t004:** Intra-group pairwise comparisons.

		Intervention Group			Control Group		
		Difference (95% CI)	*p* Value ^1^	r (95% CI)	Difference (95% CI)	*p* Value ^1^	r (95% CI)
Global MEC-35	Post-treatment–Pre-treatment	1 (1, 1)	<0.001		1 (−1.455, 1)	>0.999	
	6 months–Pre-treatment	1 (1, 1)	<0.001		1 (−0.877, 1)	>0.999	
	6 months–Post-treatment	1 (−0.298, 1)	0.426		0.929 (−1.757, 1)	>0.999	
	12 months–Pre-treatment	1 (1, 1)	<0.001	0.519 (0.406, 0.613)	1 (−0.795, 1)	>0.999	0.092 (0.003, 0.228)
	12 months–Post-treatment	1 (1, 1)	0.052		0.182 (−2.965, 1)	>0.999	
	12 months–6 months	1 (−0.426, 1)	0.315		−0.818 (−3.567, 1)	>0.999	
*Global orientation*	Post-treatment–Pre-treatment	1 (−0.018, 1)	0.353		−0.718 (−1.909, 0.473)	>0.999	
	6 months–Pre-treatment	1 (0.84, 1)	0.009		−0.643 (−1.969, 0.683)	>0.999	
	6 months–Post-treatment	0.635 (−0.422, 1)	>0.999		−0.071 (−1.536, 1)	>0.999	
	12 months–Pre-treatment	1 (0.18, 1)	0.159	0.196 (0.054, 0.36)	−1.091 (−2.746, 0.564)	>0.999	0.062 (0.002, 0.18)
	12 months–Post-treatment	0 (−1.186, 1)	>0.999		−0.636 (−2.504, 1)	>0.999	
	12 months–6 months	−0.455 (−2.014, 1)	>0.999		−0.727 (−2.06, 0.605)	>0.999	
*Temporal orientation*	Post-treatment–Pre-treatment	0.672 (−0.173, 1)	0.357		−1.077 (−2.143, −0.011)	0.75	
	6 months–Pre-treatment	1 (0.147, 1)	0.019		−1.5 (−2.899, −0.101)	0.472	
	6 months–Post-treatment	0.282 (−0.88, 1)	>0.999		−0.429 (−1.654, 0.797)	>0.999	
	12 months–Pre-treatment	0.909 (−0.165, 1)	0.628	0.151 (0.013, 0.302)	−1.364 (−2.752, 0.025)	0.615	0.123 (0.014, 0.256)
	12 months–Post-treatment	−0.455 (−1.285, 0.376)	>0.999		−0.818 (−2.135, 0.498)	>0.999	
	12 months–6 months	−0.545 (−1.883, 0.792)	>0.999		−0.545 (−1.779, 0.688)	>0.999	
*Spatial orientation*	Post-treatment–Pre-treatment	0.259 (−0.475, 0.992)	0.976		0.359 (−0.381, 1)	>0.999	
	6 months–Pre-treatment	0.847 (0.151, 1)	0.169		0.857 (−0.05, 1)	0.652	
	6 months–Post-treatment	0.494 (−0.232, 1)	>0.999		0.357 (−0.756, 1)	>0.999	
	12 months–Pre-treatment	0.818 (−0.035, 1)	0.285	0.187 (0.035, 0.332)	0.273 (−0.808, 1)	>0.999	0.039 (0, 0.196)
	12 months–Post-treatment	0.636 (−0.265, 1)	>0.999		0.182 (−0.974, 1)	>0.999	
	12 months–6 months	0.091 (−0.747, 0.929)	>0.999		−0.182 (−1.039, 0.675)	>0.999	
*Global attention and calculation*	Post-treatment–Pre-treatment	1 (1, 1)	0.001		1 (−0.851, 1)	0.984	
	6 months–Pre-treatment	1 (1, 1)	0.003		0.143 (−1.868, 1)	>0.999	
	6 months–Post-treatment	1 (−0.032, 1)	0.171		−1.286 (−3.002, 0.43)	0.55	
	12 months–Pre-treatment	1 (0.545, 1)	0.052	0.226 (0.063, 0.38)	1 (−1.462, 1)	>0.999	0.038 (0.002, 0.206)
	12 months–Post-treatment	1 (−0.637, 1)	0.862		−1 (−2.912, 0.912)	>0.999	
	12 months–6 months	−0.182 (−2.202, 1)	>0.999		0.545 (−1.333, 1)	>0.999	
*Attention*	Post-treatment–Pre-treatment	1 (1, 1)	<0.001		1 (1, 1)	<0.001	
	6 months–Pre-treatment	1 (1, 1)	<0.001		1 (1, 1)	<0.001	
	6 months–Post-treatment	1 (−0.261, 1)	0.44		−0.643 (−2.405, 1)	0.657	
	12 months–Pre-treatment	1 (1, 1)	<0.001	0.343 (0.207, 0.455)	1 (1, 1)	0.019	0.219 (0.064, 0.327)
	12 months–Post-treatment	0.818 (−0.774, 1)	>0.999		−0.636 (−2.504, 1)	>0.999	
	12 months–6 months	−0.455 (−2.157, 1)	>0.999		0.273 (−1.372, 1)	>0.999	
*Calculation*	Post-treatment–Pre-treatment	−3.362 (−5.055, −1.669)	0.002		−3.744 (−5.248, −2.239)	<0.001	
	6 months–Pre-treatment	−3.247 (−5.375, −1.119)	0.177		−4.429 (−6.358, −2.499)	<0.001	
	6 months–Post-treatment	0.565 (−0.769, 1)	>0.999		−0.571 (−1.86, 0.717)	>0.999	
	12 months–Pre-treatment	−3.364 (−5.387, −1.341)	0.008	0.272 (0.117, 0.393)	−4.273 (−6.54, −2.006)	0.005	0.244 (0.095, 0.349)
	12 months–Post-treatment	0.455 (−1.016, 1)	>0.999		−0.364 (−1.778, 1)	>0.999	
	12 months–6 months	0.182 (−1.553, 1)	>0.999		0.273 (−0.868, 1)	>0.999	
*Short-term memory*	Post-treatment–Pre-treatment	1 (1, 1)	<0.001		1 (0.653, 1)	0.017	
	6 months–Pre-treatment	1 (1, 1)	<0.001		1 (1, 1)	<0.001	
	6 months–Post-treatment	1 (0.063, 1)	0.307		1 (0.19, 1)	0.093	
	12 months–Pre-treatment	1 (1, 1)	<0.001	0.425 (0.316, 0.53)	1 (1, 1)	0.004	0.253 (0.081, 0.364)
	12 months–Post-treatment	1 (0.599, 1)	0.038		0.909 (−0.338, 1)	0.686	
	12 months–6 months	0.909 (−0.416, 1)	>0.999		0.091 (−1.266, 1)	>0.999	
*Language and global praxis*	Post-treatment–Pre-treatment	1 (1, 1)	<0.001		−0.615 (−1.718, 0.487)	0.834	
	6 months–Pre-treatment	1 (0.144, 1)	0.103		0.357 (−1.314, 1)	>0.999	
	6 months–Post-treatment	−1.624 (−3.013, −0.234)	0.214		0.857 (−0.545, 1)	>0.999	
	12 months–Pre-treatment	1 (1, 1)	<0.001	0.399 (0.266, 0.507)	0.727 (−1.148, 1)	>0.999	0.043 (0.005, 0.17)
	12 months–Post-treatment	0.636 (−0.76, 1)	>0.999		0.818 (−0.991, 1)	>0.999	
	12 months–6 months	1 (0.814, 1)	0.015		−0.727 (−2.203, 0.748)	>0.999	
*Language*	Post-treatment–Pre-treatment	1 (1, 1)	<0.001		0.41 (−0.452, 1)	>0.999	
	6 months–Pre-treatment	1 (0.447, 1)	0.015		1 (−0.044, 1)	0.174	
	6 months–Post-treatment	−0.847 (−1.766, 0.071)	0.582		0.776 (−0.332, 1)	>0.999	
	12 months–Pre-treatment	1 (1, 1)	<0.001	0.395 (0.271, 0.483)	1 (−0.407, 1)	>0.999	0.101 (0.009, 0.277)
	12 months–Post-treatment	0.273 (−0.71, 1)	>0.999		0.182 (−1.058, 1)	>0.999	
	12 months–6 months	1 (0.012, 1)	0.371		−1.091 (−2.09, −0.092)	0.34	
*Praxis*	Post-treatment–Pre-treatment	0.931 (0.113, 1)	0.271		−0.974 (−1.803, −0.146)	0.188	
	6 months–Pre-treatment	0 (−1.13, 1)	>0.999		−0.565 (−1.576, 0.446)	>0.999	
	6 months–Post-treatment	−0.847 (−1.828, 0.134)	0.729		0.353 (−0.57, 1)	>0.999	
	12 months–Pre-treatment	1 (−0.174, 1)	0.622	0.15 (0.029, 0.322)	−0.273 (−1.353, 0.808)	>0.999	0.041 (0.002, 0.169)
	12 months–Post-treatment	0.273 (−0.602, 1)	>0.999		0.545 (−0.661, 1)	>0.999	
	12 months–6 months	1 (0.184, 1)	0.165		0.182 (−0.785, 1)	>0.999	

95% CI: 95% confidence interval; MEC-35: Lobo’s cognitive mini-examination. ^1^ Significant if *p* < 0.05.

## Data Availability

Data are available from the corresponding author after reasonable request.

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
