# Peer review of "Cognitive Effects of a Cognitive Stimulation Programme on Trained Domains in Older Adults with Subjective Memory Complaints: Randomised Controlled Trial"

_ijerph, 2023, doi:10.3390/ijerph20043636_

Round 1
Reviewer 1 Report
Major concerns
1. Lots of grammatical errors, repetitive sentences, sentences without proper structure - extensive editing required. Moreover, article does not seem like it has been proofread.
2. The authors state in the Study Design section of Materials and Method that only patients who scored >= 24 on MMSE scale (even here, the authors made an error and wrote >= on the MEC-35 scale) were included in the analysis. There was no explanation provided as to why. Moreover, they should have either introduced MEC tests, and what a score of 24 and above means for it, in the introduction of the paper, or at least in the results.
3. "obtained a Lawton-Brody index ≥3; 4) reported more than 6 points on the abbreviated Goldberg anxiety scale; 5) reported ≥12 points on the abbreviated Yesavage depression questionnaire; 6) questionnaire <60 points on the Barthel Index;" - no explanations provided for what these criteria are, or why they have been selected for in the test subjects.
4. The research topic lacks novelty - a review by Hallam et al. (https://www.ncbi.nlm.nih.gov/pmc/articles/PMC8604277/) mentions two studies that already utilize CS for treatment of SMC. This study by Schultheisz et al. (https://www.ncbi.nlm.nih.gov/pmc/articles/PMC6289473/) also demonstrate the positive effects of CS on cognition, including memory. The primary novelty of the current study is that they utilize the MEC-35 scale to measure outcomes. In fact, in their discussion, the authors themselves mention - "In relation to the global cognitive effects obtained with our programme, they have also been reflected in other programmes in older adults with SMC but through multimodal and unimodal interventions [42]." and "we have identified other studies, which also report improvements in the cognitive domain of memory; some studies have achieved these results after unidomain training that only includes training of memory strategies and meta-memory in their intervention [25,46-48]."
5. The underlying statistics of Table 2 are not well explained - I did not understand the origin of the p-values. Other tables are fine.
Author Response
We would like to thank the editor for the feedback and recommendations. We have revised the paper according to the recommendations of the editor and reviewers.
Thank you very much for the reviewer's comments and suggestions, which have served to improve the manuscript. Please, see below for a point-by-point description of the changes made in the manuscript in response to them. We also enclose a revised manuscript with the changes made with change control. We appreciate these positive comments.
- Lots of grammatical errors, repetitive sentences, sentences without proper structure- extensive editing required. Moreover, article does not seem like it has been proofread.
Thank you for the appreciation. The manuscript has been corrected by a native English speaking editor, who has corrected several typos and improved writing (please, see revised text and receipt is attached).
- The authors state in the Study Design section of Materials and Method that only patients who scored >= 24 on MMSE scale (even here, the authors made an error and wrote >= on the MEC-35 scale) were included in the analysis. There was no explanation provided as to why. Moreover, they should have either introduced MEC tests, and what a score of 24 and above means for it, in the introduction of the paper, or at least in the results.
Many thanks to the reviewer for his comment, which we proceed to clarify.
The main outcome tool is MEC-35 (Lobo's Mini Cognitive Examination); in turn, this tool has been used in the inclusion criteria for screening patients without cognitive impairment, i.e., we wanted to select in our study those patients who did not present objective cognitive impairment, hence we chose those who scored 24 or more in this test. MEC-35 is the Spanish version of Folstein's MMSE, which Dr. Lobo adapted to the Spanish population.
To clarify this point we have made the following changes in the article:
1-In the Study Design section of Materials and Method, we have added that the cut-off point of 24 points or more indicates no cognitive impairment; that is, older persons with less than 24 points have cognitive impairment, according to MEC-35.
Inclusion in the study was based on the following criteria: 1) aged ≥ 65; and 2) ≥ 24 score on the MEC-35 (validated Spanish version of the Mini-Mental State Examination (MMSE)), i.e., no cognitive impairment, [33] and presentation of SMC.
2-In section 2.4. Assessment instruments, in Material and methods we have added more information about the tool, and the importance of the analysis of the different cognitive domains.
The primary variable was MEC-35 [39], which is considered one of the most widely used short cognitive tests to study cognitive capacities in Primary Care. It evaluates eight components: temporo-spatial orientation (10 points), fixation memory (3 points), attention (3 points), and calculation (5 points), short-term memory (3 points), language and praxis (11 points). Its sensitivity is 85–90% with its specificity is 69%. With this questionnaire, global cognition and cognitive functions were evaluated. Scores below 24 points could indicate dementia [33]. Unlike MMSE, MEC-35 includes a three-digit series to repeat two similarity items in reverse order, and subtraction is performed three by three from 30, instead of 7 by 7 from 100, as in the version of Folstein et al. As the number of items increases, the maximum score in this version reaches 35 points compared to 30 in the original one [39].
We considered using the Spanish version of MMSE (MEC-35) to assess global cognition and to observe if there was any change in cognitive functions. Other authors suggest further investigation as to whether the overall MMSE assessment reveals areas of concern [40]. Gómez Gallego et al. mention that the MMSE allows the rapid assessment of cognitive functions and evaluates the functions of different domains [41]. The data validity of the individual MEC items are also satisfactory (particularly with temporal orientation) [33]. In Spain, the adaptation of the MMSE carried out by Lobo et al in 1979 [42], titled MEC, is commonly employed because some items of the original version by Folstein are difficult for patients of a low cultural level, which affects the scale’s discriminative capacity [39].
- "obtained a Lawton-Brody index ≥3; 4) reported more than 6 points on the abbreviated Goldberg anxiety scale; 5) reported ≥12 points on the abbreviated Yesavage depression questionnaire; 6) questionnaire <60 points on the Barthel Index;" - no explanations provided for what these criteria are, or why they have been selected for in the test subjects.
We appreciate the reviewer's comment. We have proceeded to explain in detail the cutoff points for each of the psychological and functional tools in the instrument description section, in order to expand the information.
The reasons for choosing the psychological criteria through the Goldberg anxiety scale and the Yesavage depression scale, and excluding those participants with high anxiety and depression, is precisely to ensure that the participants can attend and participate in the cognitive stimulation sessions, as they do not present an acute psychiatric pathology that prevents them from doing so. On the other hand, the fact of considering functional aspects through the Barthel and Lawton-Brody scales helps us to determine that the participants do not have an excessive dependence that limits their access to the intervention sessions, as well as their ability to carry them out adequately.
The tools used in the inclusion criteria are as follows:
Goldberg Anxiety Subscale
Anxiety was measured by the Goldberg Anxiety Subscale, which is a subscale of the Goldberg questionnaire, with nine dichotomous response items (yes/no). An independent score is awarded for each scale with one point for an affirmative answer. The cut-off value is ≥ 4 for the anxiety subscale, which indicates “probable anxiety”. This scale has a specificity of 91% and a sensitivity of 86% [43].
Yesavage Geriatric Depression Scale the 15-point version
Level of depression was evaluated with the GDS-15, which is considered suitable for seniors in the community. Scores range from 0-15, with a total score > 5 interpreted as “probable depression”. Scoring higher than 12 would be indicative of severe depression. In older people, sensitivity is 71.8% and specificity is 78.2% for a cut-off of 5 points [44].
The Barthel Index
The BI assesses the level of independence of 10 basic ADL (BADL) [45]. The maximum score for the BI is 100, where scores higher than 60 denote low dependence with ADL and scores below 20 demonstrate high dependence with ADL. Internal consistency was 0.90, with an inter-observer reliability Kappa Index of between 0.47 and 1.00, and the inter-observer reliability Kappa Index between 0.84 and 0.97. Cronbach’s alpha was 0.90–0.9228 for the internal consistency evaluation [46].
The Lawton–Brody scale
The L-B scale assesses the degree of autonomy in eight IADL necessary for living independently in the community [45]. Scores range from 0–8 points. A score of 3 or less would be considered indicative of moderate dependence. Its sensitivity is 0.57 and its specificity is 0.92 [47]. The minimal important change of the Lawton IADL scale is around half a point. The certainty of this conclusion is reduced by variation across calculation methods [48].
- The research topic lacks novelty - a review by Hallam et al. (https://www.ncbi.nlm.nih.gov/pmc/articles/PMC8604277/) mentions two studies that already utilize CS for treatment of SMC. This study by Schultheisz et al. (https://www.ncbi.nlm.nih.gov/pmc/articles/PMC6289473/) also demonstrate the positive effects of CS on cognition, including memory. The primary novelty of the current study is that they utilize the MEC-35 scale to measure outcomes. In fact, in their discussion, the authors themselves mention - "In relation to the global cognitive effects obtained with our programme, they have also been reflected in other programmes in older adults with SMC but through multimodal and unimodal interventions [42]." and "we have identified other studies, which also report improvements in the cognitive domain of memory; some studies have achieved these results after unidomain training that only includes training of memory strategies and meta-memory in their intervention [25,46-48]."
We appreciate the commentary. We have taken into account the reviewer's comments and added them to the discussion section.
In relation to the global cognitive effects obtained through our programme, such effects have also been recorded in other programmes in older adults with SMC. In those cases, the effects were obtained through multimodal and unimodal interventions [51] and also through CS where the assessment of these benefits was conducted using the MMSE [52]. In addition, some authors advised older adults with SMQ to reduce the risk of cognitive decline through physical activity, cognitive stimulation and a healthy diet [53,54].
In our investigation, we have identified other studies, which also report improvements in the cognitive domain of memory. Some studies have achieved these results after single-domain training that only includes training of memory strategies and meta-memory in their intervention [25,58-60], whilst other authors that administered CS for treatment of SMC also found improvements in memory [52].
- The underlying statistics of Table 2 are not well explained - I did not understand the origin of the p-values. Other tables are fine.
We agree your suggestion, we added a clarification footnote in table 2: "1Anova test p values for time (within measurement time in each group) and group (between both groups) main effects and group: time interaction, Significant if p<0.05." it explain what Anova effect is each p value, hope this will be more clear. Moreover, we have improved some details in the p-values in table 1.

Reviewer 2 Report
in this paper authors performed a RTC on subjects with SMC through a protocol with Cognitive Stimulation. This is an interesting paper.
However there are some issues that need to be solved.
-There are several typos in the Abstract that, globally, seems poorly written. Please check it carefully
-Check out all the speeling and abbreviations
-It is not clear how long it is the Cognitive stimulation protocol. Please state it clearly in the abstract and throughout all the text. It could be useful also a figure depicting all the protcol (part from the CONSORT Flow Diagram)
-In the discussion I would also stress the fact that cognitive stimulation relies and improves cholinergic activity ( 10.1016/j.cobeha.2015.01.004) which is known to be responsible for memory performances ( DOI: 10.1111/ejn.13588). This aspect is fundamental in these patients, especially the ones evolving to MCI in which choliergic dysfunction has been showed to be already impaired and potentially restored by drugs (
DOI:10.3389/fnagi.2014.00254)
Author Response
We would like to thank the editor for the feedback and recommendations. We have revised the paper according to the recommendations of the editor and reviewers.
Thank you very much for the editor's comments and suggestions, which have served to improve the manuscript. Please, see below for a point-by-point description of the changes made in the manuscript in response to them. We also enclose a revised manuscript with the changes made with change control. We appreciate these positive comments.
Comments and Suggestions for Authors
in this paper authors performed a RTC on subjects with SMC through a protocol with Cognitive Stimulation. This is an interesting paper.
However there are some issues that need to be solved.
-There are several typos in the Abstract that, globally, seems poorly written. Please check it carefully
In agreement with the reviewer, we have carefully revised the Abstract and improved the wording. Moreover, the abstract section has been carefully revised by a native English speaking editor.
-Check out all the speeling and abbreviations
We appreciate the reviewer's comment and we have revised the speeling and abbreviations.
-It is not clear how long it is the Cognitive stimulation protocol. Please state it clearly in the abstract and throughout all the text. It could be useful also a figure depicting all the protocol (part from the CONSORT Flow Diagram)
In agreement with the reviewer, we have explained the different phases of the study. Relation to, this explanation is shown in Figure 1. Furthermore, we have added in the flow chart, an explanation of the origin of the participants. In addition in the “Section 2.3. Intervention” we have added one line.
2.3. Intervention
10 group sessions of 45 min, one day a week, were conducted in 5 subgroups of 26-27 persons. In total, the intervention lasted two and a half months.
|
Contact with San José Norte-Centro Health Centre |
|
1st Standardised training of assessors who conducted the first assessment |
|
CEICA approval |
|
Medical referral of cases |
|
Information posters, registration forms |
|
Telephone calls from the Health Centre to arrange appointments |
|
First assessments (baseline cognitive assessment) |
|
2nd Standardised training of assessors who administered the intervention |
|
Randomisation of intervention and control groups |
|
Treatment in the intervention groups |
|
3rd Standardised training of assessors who conducted the second assessment |
|
Second assessments (post-intervention cognitive assessment) |
|
4th Standardised training of assessors who conducted the third assessment |
|
Third assessments (6 month cognitive assessment) |
|
5th Standardised training of assessors who conducted the fourth assessment |
|
Fourth assessments (12 month cognitive assessment) |
Figure 1. Protocol for the different phases of the study.
-In the discussion I would also stress the fact that cognitive stimulation relies and improves cholinergic activity (10.1016/j.cobeha.2015.01.004) which is known to be responsible for memory performances ( DOI: 10.1111/ejn.13588). This aspect is fundamental in these patients, especially the ones evolving to MCI in which choliergic dysfunction has been showed to be already impaired and potentially restored by drugs (DOI:10.3389/fnagi.2014.00254)
We appreciate the reviewer's comment and we proceed to include the comment in the discussion.
Moreover, cognitive stimulation relies and improves cholinergic activity [77], which is known to be responsible for memory performance [78]. This aspect is fundamental in these patients, especially those evolving to MCI in which cholinergic dysfunction has been shown to be already impaired and potentially restored by drugs [79].

Round 2
Reviewer 1 Report
Thank you to the author's for addressing all of my concerns.